# Permissive parenting of the dog associates with dog overweight in a survey among 2,303 Dutch dog owners

Ineke R. van Herwijnen[1]*, Ronald J. Corbee[2‡], Nienke Endenburg[3‡], Bonne Beerda[1],
Joanne A. M. van der Borg[3]

1 Department of Animal Sciences, Behavioural Ecology Group, Wageningen University and Research,
Wageningen, The Netherlands, 2 Department of Clinical Sciences of Companion Animals, Faculty of
Veterinary Medicine, Utrecht University, Utrecht, The Netherlands, 3 Department of Population Health, Unit
Animals in Sciences and Society, Faculty of Veterinary Medicine, Utrecht University, Utrecht, The
Netherlands

☉ These authors contributed equally to this work.
‡ These authors also contributed equally to this work.
* ineke.vanherwijnen@wur.nl

org/10.1371/journal.pone.0237429

Instituto de Biologia Molecular e Celular,
PORTUGAL

**Data Availability Statement:** All relevant data are
within the paper and its Supporting Information
files.

## Abstract

Overweight/obese dogs are at increased risk of health issues and it is up to the dog owner to
uphold successful weight management. In children, overweight relates to their parent's per-
missive style of parenting. We predicted that permissive dog-directed parenting likewise
associates with a dog being overweight (including obesity). If styles in parenting dogs indeed
associate with a dog's overweight, these may provide action points for effective weight man-
agement. For 2,303 Dutch dog owners, answers on their dog's (nine-point scale) body con-
dition scores were compared to ways of parenting the dog. We used an adapted version of
the 32-item Parenting Styles and Dimensions Questionnaire and compared the distributions
of dog counts across aggregated body condition score categories of underweight (scores
one to three), healthy-weight (scores four and five) and overweight/obese (scores six to
nine) with Chi-square tests across the quartiles of a given parenting style. Overweight/
obese dogs were overrepresented in the quartile of dog owners with the highest level of per-
missive parenting, which is in line with findings on parenting styles and overweight/obesity in
children. Supplementary logistic regression analyses on the likelihood of dogs being over-
weight/obese (i.e. having a body condition score of six or higher) confirmed the importance
of parenting and identified the risk factors of dogs having little exercise, being of older age,
neutered or owned by someone with lower level education. Our results indicate that strate-
gies to promote proper weight management in dogs could benefit from addressing espe-
cially a dog owner's permissiveness in parenting his/her dog.

**Funding:** The authors received no specific funding for this work.

**Competing interests:** The authors have declared that no competing interests exist.

## Introduction

Overweight in dogs reduces quality of life [1, 2] by causing serious health problems such as musculoskeletal disorders, neoplasia and disturbances of normal endocrine functions [3, 4]. One way of assessing overweight is by means of body condition score measurement, which is a non-invasive method that has been validated with dual-energy X-ray absorptiometry, bioelectrical impedance and thoracic radiography of subcutaneous fat [5–7]. Body condition score measurement combines visual inspection with palpation of the dog. On a nine-point scale the body condition score measurement expresses the broader categories of underweight (scores one to three), healthy-weight (scores four and five) and overweight/obese (scores six to nine), with six and seven indicating overweight and eight and nine indicating obesity [8–10].

Overweight (including obesity) in dogs has been studied for many years, covering aspects of its epidemiology, pathophysiology, management and comorbidity in dog and owner [4, 11–16]. Also, owner aspects of feeding and exercising the dog have been researched. Findings indicate owner effects on the dog's weight. For instance, overweight dogs, more so than healthy-weight dogs, were found to be fed semi-moist foods [13], homemade foods, table scraps such as bread, meat, pasta, sausage [17] and snacks/treats [13, 18, 19]. Next to feeding differences, overweight dogs were found to have lower exercise levels [11, 18, 19]. If dog owners vary considerably in how they feed and exercise their dog [11, 13], how they view and relate to it may be key to this. For instance, owners of overweight dogs were more likely to see them 'as a baby' and allow them to sleep on the bed [20]. Also, owners of overweight dogs tended to value the dog less for exercise, work and/or protection purposes but spoke to it more and on a larger variety of subjects [21].

Differences in how owners see and treat their dog can be studied through the recently discovered dog-directed parenting styles [22, 23] and in humans associations exist between parenting styles and a child's overweight, including obesity [24]. Parenting styles encompass the overarching 'emotional climate' in the relationship between a care provider such as a dog owner and a care receiver such as a dog [23]. This emotional climate is characterized by variation in dimensions of responsiveness and demandingness [25]. Responsiveness is about recognising the needs and emotions of the care receiver. Demandingness is about providing boundaries, exerting control and monitoring of behaviour and performance [25]. Scoring high on both responsiveness and demandingness is described as an authoritative style of parenting, which is considered as balanced and most optimal [25]. The permissive style is characterized by high levels of responsiveness and low levels of demandingness, reflecting unbalanced parenting. Another form of unbalanced parenting is the authoritarian style with high levels of demandingness and low levels of responsiveness [25]. Similar styles of dog-directed parenting reflect the differences in responsiveness and demandingness with added reflections of correction, intrinsic value or training orientations towards a dog's parenting [23].

Findings on child-directed parenting suggest that permissive parenting promotes overweight/obesity [24] and in the owner-dog relationship this may work for example by an eagerness to please the dog with food (high responsiveness) without a counterbalancing demandingness regarding exercise and, ultimately, body condition scores. This study aims to test a presumed influence of dog owners on dog overweight/obesity, focussing on parenting styles directed at the dog. Revealing associations between dog-directed parenting styles and body condition scores in dogs may point out new weight management strategies as to promote healthier dogs.

## Methods

### General approach and ethical considerations

We tested how dog-directed parenting styles associated with dog weight status, as reflected in body condition scores. Our sample consisted of dog owners recruited via advertising online and in hardcopy dog magazines, with press releases in national and regional news channels, including newspapers. For responding to the survey, the participants were not invited to a research location, but presumedly responded to our web-based survey on a private computer, at home or elsewhere, hence minimizing social pressure to provide certain answers. Anyone owning a dog and caring for it at for least half of the time was eligible to participate in the study. Caring for the dog was described as walking/training the dog, feeding the dog and performing other husbandry tasks. Additional criteria, such as fluency in Dutch were not made. The aim of the survey was explained to participants generally, without mentioning the specific element of a dog's weight status. Specifically, the aim was explained as 'each dog owner having his/her own view on the dog and on raising it; and Wageningen University and Research wishing to study people's views on raising the dog and possible outcomes thereof for the dog and the relationship with the dog'.

The survey was taken once, we did not contact the participants for a second survey, did not include questions that were psychologically burdening, that is psychologically difficult to bear such as on mental health status, meaning it was unlikely to interfere significantly with normal daily life. This exempted the study from review by our ethics committee, according to the guidelines of Wageningen University Medical Ethics Review Committee (Medisch Ethische Toetsingscommissie van Wageningen University, METC-WU).

### Survey

The web-based survey was in Dutch and gathered information on dog and owner characteristics such as gender, age and dog-directed parenting styles between August 2017 and September 2018. Sample sizes were not estimated through power calculations before making the survey available online and we did not predetermine a survey period. However, 95% of the data was collected in the first three months after starting the survey advertisement. Parenting styles were measured with 32 parenting style items on a five-point Likert scale, rating the likelihood of scenarios occurring as never (score zero), nearly never (one), neutral (defined as about half of the time, two), nearly always (three) and always (four). These items were based on the Parenting Styles and Dimensions Questionnaire (32-PSDQ) commonly used with children [26] and transformed for use with dogs [23]. Here we used the 32 items measuring on the three 'original' parenting styles of authoritative, authoritarian and permissive parenting, as to facilitate comparisons of our study outcomes with similar studies done with children. We did an additional analysis using the twenty items that make up the previously determined dog-directed parenting styles (DD-PSDQ) of authoritative-training orientated, authoritative-intrinsic value orientated and authoritarian-correction orientated parenting and which largely overlap the 32 items, but for the addition of two items as described in the earlier mentioned study by Van Herwijnen et al. [23] and S1 Appendix lists these parenting style questions. For each parenting style the scores were calculated by combining item scores into a percentage of the theoretical maximum of 100%. Note that we did not study associations with the parenting style of uninvolvedness as the original child-directed measurement tools did not measure this style [27].

Body condition scores were measured on a scale of one to nine (BCS 1–9, 9 inclusive). Scores one to three represented underweight (BCS 1–3), four and five healthy-weight (BCS

4–5), six and seven overweight (BCS 6–7) and eight and nine obesity (BCS 8–9). Dog owners were presented with pictures and descriptions of body condition scores as propagated by the World Small Animal Veterinary Association (WSAVA) in their nutrition toolkit, as for the owners to determine their dog's weight status (see S1 Appendix for details). Thus, they assessed their dog's body condition score themselves, based on this text/image-based instruction.

The survey included six general questions about the participating owners and their dogs and we used this information to test for modifiers of the relationship between dog-directed parenting and the likelihood of dogs being overweight/obese, as well as explaining variation in the latter. The questions about the dogs involved: how long they were exercised on a typical day (<30 minutes, 30–59 minutes, 60–89 minutes, ≥90 minutes), their age (1, 2, 3, 4, 5, 6, 7, 8, 9, ≥10 years) and sex/neutering status (intact males, neutered males, intact females, neutered females). The participating dog owners reported their level of education (elementary school level only, high school level, undergraduate level, master degree or higher), age (<25 years, 25–34 years, 35–44 years, 45–54 years, 55–64 years, ≥65 years) and gender (male/female).

The answering of questions in the survey was not mandatory and as a consequence unanswered questions represent missing values. From the received survey records a total of 732 records was removed. Reasons for removal were a record being a replicate of another survey record, containing more than 50% empty values, and/or empty values that prevented us from calculating parenting style scores or body condition scores (52 surveys). Thus, a total number of 2,303 surveys was available for our statistical analyses.

## Statistical analyses

The survey data were checked for replicates via the combination of the dog's name and postal code, which we used to label the data recognisably, while allowing participants to participate anonymously. We calculated if dog counts for the body condition score categories underweight (scores one to three), healthy-weight (scores four and five) and overweight/obese (scores six to nine) were spread differently over the first, second, third and fourth quartile of the parenting style scores. These quartiles were based on the actual data distribution and range, not the theoretical value range of 0–100%.

For the classification of body condition scores into these three categories, we followed earlier studies using the same range of body condition scores [8, 9, 10] and we opted for this approach as the dogs' weights were not evenly spread over all body condition score categories, resulting in low counts in some categories. We did not presume normal distribution and therefore present descriptive data as medians (ranges). Our main interest was to study if overweight/obesity in dogs associates with parenting styles, following findings in child-directed studies, and we used Chi-square tests to test for a relationship between the categorical variables body condition score and parenting style. Chi-square tests outcomes were evaluated with the threshold of significance set at P<0.001. This instead of P<0.05 to separate the more biologically meaningful associations from weaker ones that reached significance by the large sample size ($N$ = 2,303). With the Chi-square tests, we present standardized residuals to identify the cells with the largest contribution to the Chi-square test results. We mark residuals |>2| bold as this threshold is commonly accepted as a sufficiently large deviation between observed and expected values [28]. To check for possible overlap between the parenting styles, we calculated Spearman's rank correlations as to provide insight into the basic characteristics of our study sample. Statistical analyses were performed with GenStat (18th edition) software.

Our survey was designed primarily to establish the relation between a dog's body condition score and its owner's dog-directed parenting style, but supplementary to this we evaluated the

(modifying) effects of six basic characteristics of owners or dogs. For this we used logistic regressions on the dog's overweight/obese body condition scores (BCS 6–9) as the binary response variate (1; versus BCS 1–5 as 0). The explanatory dog variables were the amount of daily exercise (four levels), sex/neutering status (four categories) and age (ten levels). The owner variables were educational level (four levels), age (six levels) and gender (two categories). Explanatory variables were expressed as (continuous) co-variates, but for sex/neutering status and gender. The six different parenting style scores, expressed as a percentage of theoretical maximum, were included as co-variates and analysed one by one. Two types of regression models were used. First, we analysed the twelve explanatory variables singly. Next, we ran regressions with one parenting style plus six owner/dog variables, including two-way interactions with the parenting style, and reduced the models by stepwise backwards elimination, though maintaining parenting style as a main effect. The least significant variable that was not marginal to another was eliminated from the model using a threshold of $P<0.05$ (Wald tests). The logistic regression predicted mean fractions of dogs with overweight/obesity (±s.e.) are presented for the range of the 50% middle values of parenting styles, meaning the two central quartiles; i.e. the range of common values. This, to enable meaningful comparisons between parenting styles regarding the relationship strengths with overweight/obesity in dogs.

## Results

### Participants and their dogs

The study sample included 2,303 dog owners who filled out the online survey between August 2017 and September 2018, excluding 52 surveys that had a missing value for the dog's body condition score. The answering of questions in the survey was not mandatory and unanswered questions resulted in missing values. Table 1 presents the sample distribution of owner-reported body condition scores (BCS), of dogs being underweight (BCS 1–3), healthy-weight (BCS 4–5) and overweight/obese (BCS 6–9) and of parenting style median scores and ranges.

**Table 1. Descriptive study sample data on dog Body Condition Score (BCS) and parenting styles.**

| Dog body condition score (BCS) | | | |
|---|---|---|---|
| **BCS** | **% (total)** | **BCS category (scores)** | **% (total)** |
| BCS 1 | <1% ($N = 2$) | Underweight (BCS 1–3) | 18.5% ($N = 427$) |
| BCS 2 | <1% ($N = 15$) | | |
| BCS 3 | 18% ($N = 410$) | | |
| BCS 4 | 24% ($N = 554$) | Healthy-weight (BCS 4–5) | 75% ($N = 1,727$) |
| BCS 5 | 51% ($N = 1,173$) | | |
| BCS 6 | 4% ($N = 100$) | Overweight/obese (BCS 6–9) | 6.5% ($N = 149$) |
| BCS 7 | 2% ($N = 40$) | | |
| BCS 8 | <1% ($N = 5$) | | |
| BCS 9 | <1% ($N = 4$) | | |
| **Parenting style score medians (ranges) for original 32-PSDQ and for DD-PSDQ** | | | |
| Parenting style | | Median (range) | |
| Authoritative style | | 75.0% (20.0–100%) | |
| Authoritarian style | | 22.9% (0–83.3%) | |
| Permissive style | | 25.0% (0–91.7%) | |
| Authoritative-training orientated style | | 87.5% (8.3–100%) | |
| Authoritarian-correction orientated style | | 23.0% (0–84.4%) | |
| Authoritative-intrinsic value orientated style | | 65.0% (4.2–100%) | |

Over all dogs, the body condition score median was 5 (range 1–9), with low counts in the highest and lowest BCS categories. The dogs were of varying breeds and 60% ($N = 1,372$) had a pedigree ($N = 908$ non-pedigree) ($N = 23$ missing values). Most dogs were walked for $\geq 90$ minutes daily (71%, $N = 1,617$). A walking duration of 60–89 minutes was reported for 23% of dogs ($N = 524$), of 30–59 minutes for 6% ($N = 136$) and of <30 minutes for <1% ($N = 16$) (10 missing values). Intact dogs made up 51% ($N = 1,125$, with 693 males and 432 females), neutered dogs 49% ($N = 1,071$, with 463 males and 608 females) (107 missing values). Weights of the dogs distributed as 3% ($N = 79$) <5 kilos, 12% ($N = 284$) 5–10 kilos, 20% ($N = 457$), 11–20 kilos, 30% ($N = 691$) 21–30 kilos, 23% ($N = 516$) 31–40 kilos, 8% ($N = 191$) 41–50 kilos and 4% ($N = 80$) >50 kilos (5 missing values). The majority of the participants was female (86%, $N = 1,971$; male: 14%, $N = 317$) (15 missing values). Elementary school level only was obtained by 1% ($N = 23$), high school level by 20% ($N = 447$), undergraduate level by 67% ($N = 1,535$) and master degree or higher by 12% ($N = 284$) (14 missing values). The participants' age distribution comprised that 7% ($N = 162$) was <25 years, 21% ($N = 462$) 25–34, 19% ($N = 421$) 35–44, 30% ($N = 681$) 45–54, 18% ($N = 392$) 55–64, and 5% ($N = 113$) $\geq 65$ years or older (72 missing values).

We checked for overlap between the three original parenting styles of authoritative, authoritarian and permissive parenting (32-PSDQ), between the DD-PSDQ and between the 32-PSDQ and DD-PSDQ and found correlations in line with the earlier study done by us. The Spearman's rank correlations are provided in S1 Table.

## Parenting style scores and body condition score categories

We used Chi-squares to test if the dog count distribution across the three main body condition score categories (underweight, healthy-weight and overweight/obese) differed between the four levels of each of the dog-directed parenting styles of the first, second, third and fourth quartile of parenting style scores. The association between body condition score and parenting was significant at the level of P<0.001 only for permissive parenting. Here, the number of overweight/obese dogs was higher than expected for the higher (fourth) quartile of permissive style scores and lower than expected for the lower (first) quartile of permissive style scores ($\chi^2$ = 33.8, P<0.001, df = 6, $N = 2,303$; see Table 2 and S2 Table presents all counts for all parenting styles).

## Modifiers of relationships between parenting style and overweight/obesity

The relationships between dog-directed parenting styles and overweight/obesity in dogs were investigated in more detail by means of logistic regressions (BCS 6–9 as 1 versus scores 1–5 as 0), including the testing of six candidate modifiers that characterized dogs or owners (dog's

**Table 2. Counts of underweight, healthy-weight, overweight/obese dog body condition scores per quartiles of an owner's permissive parenting style scores.**

| | Underweight (BCS 1–3) | Healthy-weight (BCS 4–5) | Overweight/obese (BCS 6–9) |
|---|---|---|---|
| **Permissive style score 0–18.75%** | 121 (1.62) | 441 (0.51) | **20 (-3.44)** |
| **Permissive style score 18.76–25.00%** | 138 (1.72) | 488 (-1.21) | 40 (-0.58) |
| **Permissive style score 25.01–35.00%** | 87 (-1.64) | 420 (1.80) | 32 (-0.57) |
| **Permissive style score 35.01–91.67%** | 81 (-1.89) | 378 (-1.03) | **57 (4.80)** |

A dog's body condition score (BCS) being underweight (grouping BCS 1–3), healthy-weight (BCS 4–5) or overweight/obese (BCS 6–9) was calculated to fall into an owner's first to fourth quartile of scores for each parenting style. These quartiles were based on the actual data distribution, not the theoretical value range of 0–100%. Chi-square tests for these frequencies were significant only for permissive parenting and we present counts (residuals), marking in bold the observed counts that deviate (residual |>2|) from expected counts ($\chi^2$ = 33.8, P<0.001, df = 6, $N = 2,303$; all other P>0.001).

exercise time, age, sex/neutering status, and owner's educational level, age and gender). These characteristics were furthermore evaluated for explaining variation in the likelihood of dogs being overweight/obese. Regressions with single explanatory variables confirmed the earlier findings for permissive parenting (predicted mean ±s.e. fractions of dogs with overweight/obesity of 0.05±0.01 at 19% permissive parenting and 0.07± 0.01 at 35%, which represents the range of 50% middle values, P<0.001). Similarly, the authoritarian styles were related directly with overweight/obesity (authoritarian, 0.05±0.01 at 15% and 0.07±0.01 at 33%, P = 0.006; authoritarian-correction, 0.06±0.01 at 13% and 0.07±0.01 at 31%, P = 0.007), whereas the parenting style authoritative-training related inversely (0.07±0.01 at 75% and 0.05±0.01 at 92%, P<0.001). However, this latter effect was a trend only (P = 0.076) when owner/dog characteristics were added to the statistical model and for an overview of all results see S3 Table.

Risks of overweight/obesity in dogs were little exercise of the dog, old age, neutering, and lower education in owners. For daily exercise time the predicted fractions of overweight/obese dogs ranged from 0.20±0.05 (<30 minutes) to 0.05±0.01 (≥90 minutes, P<0.001 for regression with a single explanatory variable). For dog age the range was from 0.04±0.00 (1–2 years) to 0.14±0.02 (≥10 years, P<0.001). Neutered dogs had higher predicted fractions (0.09±0.01 for both males and females) than intact dogs (0.05±0.01 for females and 0.03±0.01 for males, with pairwise differences of t-probability P<0.001 for intact males compared to neutered males and females and P<0.05 for intact females compared to neutered males and females). For the owner's levels of education the fractions ranged from 0.12±0.03 for elementary school only to 0.04±0.01 for master degree or higher (P = 0.006). Owner age did not relate to the fraction of dogs with overweight/obesity (P = 0.45), although women seemed more likely to have a dog with overweight/obesity than men (0.07±0.01 versus 0.04±0.01, P = 0.035).

Relationships between parenting styles and overweight/obesity (BCS 6–9) may be modified by the aforementioned characteristics of owners and dogs, which was tested with regression models of one parenting style and the six owner/dog characteristics, including two-way interactions with parenting style. Models were then reduced by stepwise backward elimination (for the results see S3 Table). For permissive parenting there was a significant interaction effect identifying the dog's sex/neutering status as a modifier (Wald test P = 0.044, from 0.03±0.01 to 0.06±0.01 for intact females, 0.03±0.01 to 0.03±0.01 for intact males, 0.06±0.01 to 0.07±0.01 for neutered females, from 0.04±0.01 to 0.08±0.01 for neutered males over the range of 19–35% permissive parenting). Using intact males as the reference for the pairwise comparisons for sex/neutering status (as these intact males had the lowest predicted means) a significant contrast was found between permissive parenting x intact male and permissive parenting x intact female (t-probability of pairwise differences P = 0.038).

## Discussion

Overweight/obese dogs are at increased risk of poor health and it is important that dog owners adopt effective long-term weight management as part of a healthy owner-dog relationship. Weight management of the dog could be one expression of dog-directed parenting, similarly to child-directed parenting associating with a child's weight. Indeed, we found that dog owners who reported to be strongly permissive in their dog-directed parenting were more likely to own a dog that was overweight/obese (i.e. body condition scores above six on a scale of one to nine).

A relation between permissive dog-directed parenting and higher weight in dogs corresponds with findings on permissive child-directed parenting relating to higher weight in children, as concluded from a review of 23 cross-sectional studies, seven longitudinal and one randomized control trial [24]. As an example, a regression coefficient of 0.35 was found for

permissiveness and the child's higher weight (P<0.05), thus explaining twelve percent of variation in child weight after controlling for factors such as parent affect, parent weight and child temperament [29]. This in a survey on 718 parents, of which 240 parented permissively [29]. Some studies reported the permissive style to even double a child's chances on overweight [30]. However, other studies report mixed results for the associations between a child's weight and parenting styles in general [31, 32]. This indicates a need for more causal evidence [33] as the influence of the parenting environment on a child's weight status is complex [34], though probably existent with child-directed parenting styles logically relating to several child-directed feeding/exercise behaviours [35–41]. Specifically, parental permissiveness combined with poorer quality of children's diets, less monitoring of food intake, less meal-time structure, fewer food rules [35–37], and with higher levels of sedentary behaviour in the form of watching television [38]. Contrastingly, parental authoritativeness combined with more rules in place regarding 'television-time' [39, 40]. Finally, parental demandingness, which is low in permissive parenting, combined with a child's higher perceived abilities to exercise [41], increasing the chances of sufficient exercise being a part of daily routines.

There is a proposed mechanism through which parenting styles affect weight statuses long-term, at least in children [24, 42–44]. Parenting styles are thought to affect weight status in children through influencing the child's mechanisms of self-regulation/control [24, 42–44]. Self-regulation/control are regulatory responses that need to be practiced and developed during childhood and adolescence [42, 44]. The development is facilitated by authoritative parenting which combines parental demandingness and responsiveness. Firstly, appropriate parental demandingness ensures that the child is given tasks it can fulfil and this is combined by the parent monitoring/controlling the outcomes of these tasks. This allows the child to practice the tasks, which could be about eating vegetables or being active instead of sitting behind the television or tablet. When the child performs the tasks successfully and repeatedly, the proper food/exercise habits become internalized and are thus shown long-term. Secondly, the development of self-regulation/control is facilitated by responsiveness. Responsiveness ensures that a child's innate hunger/satiation signals are not overruled by parental constraints [31, 44]. An example of parental constraint is obligatory finishing a meal when the child feels satiated already. The consequence may be diminished satiation recognition over time, leading to overeating by lack of self-regulation long-term. Assumingly, the balance in parental demandingness *and* responsiveness allows children to develop appropriate self-regulated/controlled habits that benefit weight management.

Internalized food/exercise habits and self-regulation/control as underlying factors of healthy food/exercise habits may be of lesser importance to dogs than children. A child, when moving into adulthood, will increasingly form habits and self-regulate/control food/exercise behaviour as it becomes more and more independent [45, 46]. In contrast to this, most dogs remain dependent on the owner throughout life. For instance, the owner's regulation of food provision will determine the dog's food intake more than a dog's self-regulation if access to food is not freely provided to the dog. Consequently, demandingness in dog owners may not have similar long-term effects on dogs as demanding parents on children. Overruling a dog's innate hunger/satiation signals or suppressing a dog's internal exercise motivation will not affect an adult dog's weight status if the dog's food/exercise habits are still controlled by the owner during the dog's adult life. How exactly ways of parenting may relate to a dog's weight is therefore less easy to predict. We find some evidence for an increased risk of overweight/obesity in authoritarian parented dogs and particularly permissively parented dogs (and this in contrast to those parented authoritatively). However, the findings were not consistent across different statistical analyses and require further validation.

Notwithstanding the need for further validation, higher levels of responsiveness without demandingness, as characteristic to permissive parenting was here found to relate to higher dog weight. Permissiveness may play an adverse role in a dog's weight maintenance in similar ways as parent-child situations [35–37]. In the owner-dog situation also, a lack of demandingness could result in providing low quality diets to the dogs, a lack of household rules on food giving and a sedentary/minimal exercise life style. In turn, high responsiveness may make dog-owners vulnerable to feed the dog according to its requests for food giving, resulting in more frequent snack/treat giving [19]. The increased number of overweight/obese dogs in the quartile of our most permissive owners may be explained in this way. Future studies have to unravel how permissive dog-directed parenting actually expresses in the feeding and exercising of dogs in the way that builds up unwanted levels of adipose tissue.

How parental demandingness may or may not factor in a dog's weight status remains open to further studying. Particularly the DD-PSDQ style of authoritative-training orientation seems of interest as a style that may benefit a healthy weight in dogs. We found an indication for a possibly protective role in a logistic regression model. However, this remained a trend only when other explanatory variables were added to the statistical model and our Chi-square tests did not reveal this relation. We emphasize the characteristics of our study sample which was a self-selected convenience sample. Consequently, participating dog owners, who were mainly female, educated and willing to make the effort to participate in research, will not reflect a general population of dog owners. The relatively high percentage of owners walking their dogs for 90 minutes or more daily indicates a willingness to invest in the dog through exercise. Likely, our study sample is willing to invest in the dog in more ways than exercising it for longer durations. Such compassion for the dog may come with only rarely displaying extremely weak parental demandingness and/or responsiveness. This is seen for instance in the high levels of authoritative parenting in the sample. We searched for demographics of the Dutch dog owner population but were unable to find statistics that could be used for comparison with our study group. Recruiting participants with extremely weak demandingness/responsiveness levels is complex, but necessary for a complete picture of dog-directed parenting styles and possible consequences thereof. Our study sample was further characterized by dogs mainly being reported to have a healthy weight. Our found percentage of 75% healthy-weight is higher than the 64% of dogs reported to be normal weight in a study on 3,185 European dog owners (underweight: 19% versus 14%; overweight/obese: 7% versus 22%) as measured on a body condition scores ranging from one to five [47]. Additionally, a limitation of our study is it being based on dog owner self-reports. We supported valid body condition score assessments by providing both clear textual descriptions as well as graphics on the nine possible body condition scores, but owner reports may have been inaccurate or biased towards reporting lower body condition scores. Previous studies indicated that dog owners underreport their dog's weight. For instance, 44% of 680 dogs' body condition scores were misinterpreted by their owners, as compared to veterinarians, with 77% of the discrepancies being underestimations [48]. Seemingly, using a body condition score measurement does not improve underestimation, as in another study 65% of 110 owners incorrectly estimated the dog's weight status regardless whether they did or did not use a body condition score measurement [49]. Underestimation and underreporting would particularly affect our study when entangled with parenting styles. This would be an interesting topic to address in future studies on a dog's weight and parenting styles. High demandingness could come with considering it important to have a 'perfect weight dog', thus influencing reporting of body condition scores through a mechanism of social desirability. Follow-up studies could address this issue by involving expert rates on the dogs' body condition scores. Furthermore, hypotheses-driven experimental study set ups are needed to confirm causality of our here found associative

results. We also suggest to investigate a possible entanglement between permissiveness in dog-directed parenting and other variables that may affect a dog's weight, such as the decision to neuter a dog or not. Effects of permissiveness on weight may vary depending on a dog's sex and neutering status, as indicated by our finding of permissiveness particularly affecting predicted means of being overweight/obese in intact female dogs in the logistic regression model.

Finally, future studies could investigate if addressing permissiveness in dog owners may help them to resist a dog's food giving requests, adhere to proper exercise schemes and consequently facilitate a dog's healthy weight. This, as especially longer-term weight management in dogs seems challenging [1, 50, 51]. Therefore, such science could assist veterinarians in tailoring their advice to an individual owner. Already, by indicating alternative ways for being responsive than giving in to feeding requests, veterinarians can help owners to obtain and maintain healthy-weight in their dog. The alternatives would allow the owner to express responsiveness, but in a healthy way: playful active interactions, allowing the dog to do so-called 'nose work', search games and the like. Dogs are known to enjoy working with the owner and for daily food rations [52], a concept generally known as contrafreeloading [53]. The strong parental responsiveness in permissive dog owners may coincide with a strong willingness to invest in their dog. Weight management strategies could tap into this while recognising such owners' particular pitfalls.

## Conclusion

Overweight/obese dogs were overrepresented in the group of dog owners that scored highest for permissive dog-directed parenting. The combination of weak parental demandingness with strong responsiveness seems to hamper effective weigh management in dogs, challenging us to look at how to optimally support such dog owners when we ask them to feed the dog less calories and up their exercise time. Promoting the appropriate levels of demandingness and responsiveness towards dogs may help dog owners persist in those feeding/exercise behaviours that benefit a dog's healthy weight, quality of life and longevity.

## Supporting information

**S1 Table. Spearman rank correlations between the parenting styles directed at the dog.** Spearman rank correlations were calculated between the original 32-PSDQ parenting styles of authoritative, authoritarian and permissive parenting and the DD-PSDQ parenting styles determined in the previous study by Van Herwijnen et al., 2018 of authoritative-training orientated, authoritative-intrinsic value orientated and authoritarian-correction orientated parenting ($N$ = 2,303, P<0.001 for all).
(PDF)

**S2 Table. Counts of underweight, healthy-weight, overweight/obese dog body condition scores per quartile of an owner's parenting style scores.** A dog's body condition score (BCS) being underweight (grouping body condition scores one to three), healthy-weight (score four and five) or overweight/obese (score six to nine) was calculated to fall into an owner's first to fourth quartile of parenting style scores for each of the three parenting styles of authoritative, authoritarian and permissive parenting and the additionally analysed specific styles of authoritative-training orientated, authoritative-intrinsic value and authoritarian-correction orientated parenting. Chi-square tests for these frequencies were significant only for permissive parenting ($\chi^2$ = 33.8, P<0.001, df = 6, $N$ = 2,303; all other P>0.001).
(PDF)

**S3 Table. Predicted mean fractions of overweight/obese dogs from logistic regressions with parenting styles and other explanatory variables.** Dog overweight's (including obesity) association with parenting styles was tested with logistic regression. The binary response variate of overweight discriminated Body Condition Scores (BCS) 6–9 (score 1) from BCS 0–5 (0). Regression models with single parenting styles only were extended with other possibly weight-explaining variables (exercise duration, dog age, dog sex/neutering status, owner educational level, owner age, owner gender), including two-way interactions with the styles. Full models with these seven explanatory variables and six interaction terms, were reduced by stepwise backwards elimination using Wald statistics, omitting the least significant term that was not marginal to another term at a threshold of P<0.05. For the pairwise comparisons for sex/neutering status we chose the lowest predicted mean of intact males as the reference. We present predicted mean fractions (±s.e.) for significant main effects/interactions (Wald test P<0.05) for the range of the 50% middle values (the two central quartiles; i.e. the range of common values) of each parenting style. Differences were significant for authoritarian (correction orientated), authoritative-training orientated and permissive parenting (P<0.05) in the single factor model and for authoritarian (correction orientated) and permissive parenting (interaction with dog sex/neutering status in the backwards elimination model (P<0.05)).
(PDF)

**S1 Appendix. Questionnaire items.** For this survey-based research we tested the association between a dog owner's parenting style directed at the dog and the dog's body condition scores. Parenting style questionnaire items follow Van Herwijnen et al., 2018. Body condition scores were measured with pictures and descriptions of body condition scores-chart as propagated by the World Small Animal Veterinary Association (WSAVA) in their nutrition toolkit, and follow: https://www.wsava.org/WSAVA/media/Arpita-and-Emma-editorial/Body-Condition-Score-Dog.pdf.
(PDF)

**S1 Data.**
(XLSX)

## Author Contributions

**Conceptualization:** Ineke R. van Herwijnen, Bonne Beerda, Joanne A. M. van der Borg.

**Formal analysis:** Ineke R. van Herwijnen, Bonne Beerda.

**Methodology:** Ineke R. van Herwijnen, Ronald J. Corbee, Bonne Beerda.

**Supervision:** Ineke R. van Herwijnen, Bonne Beerda, Joanne A. M. van der Borg.

**Writing – original draft:** Ineke R. van Herwijnen.

**Writing – review & editing:** Ineke R. van Herwijnen, Ronald J. Corbee, Nienke Endenburg, Bonne Beerda, Joanne A. M. van der Borg.

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
