## [Decision Letter · Decision Letter 0]

15 Jan 2020

PONE-D-19-29007

Permissive parenting of the dog associates with dog overweight in a survey among 2,303 Dutch dog owners

PLOS ONE

Dear Ir. Herwijnen,

Thank you for submitting your manuscript to PLOS ONE. After careful consideration, we feel that it has merit but does not fully meet PLOS ONE’s publication criteria as it currently stands. Therefore, we invite you to submit a revised version of the manuscript that addresses the points raised during the review process.

Both reviewers are of the opinion that the paper reports interesting and relevant data, but that some substantial revisions are required. This includes a reconsideration of some of the statistical analysis. You will find the detailed review reports at the end of this email. Please take all aspects into consideration and respond to them.

We would appreciate receiving your revised manuscript by Feb 29 2020 11:59PM. To enhance the reproducibility of your results, we recommend that if applicable you deposit your laboratory protocols in protocols.io, where a protocol can be assigned its own identifier (DOI) such that it can be cited independently in the future. For instructions see: http://journals.plos.org/plosone/s/submission-guidelines#loc-laboratory-protocols

We look forward to receiving your revised manuscript.

Kind regards,

I Anna S Olsson, Ph.D.

Academic Editor

PLOS ONE

Journal Requirements:

Reviewers' comments:

Reviewer's Responses to Questions

**Comments to the Author**

1. Is the manuscript technically sound, and do the data support the conclusions?

Reviewer #1: Partly

Reviewer #2: Partly

2. Has the statistical analysis been performed appropriately and rigorously? 

Reviewer #1: No

Reviewer #2: N/A

3. Have the authors made all data underlying the findings in their manuscript fully available?

Reviewer #1: No

Reviewer #2: Yes

4. Is the manuscript presented in an intelligible fashion and written in standard English?

Reviewer #1: Yes

Reviewer #2: Yes

5. Review Comments to the Author

Reviewer #1: The authors studied the associations between the body condition of dogs and their owners’ parenting styles. It is very interesting to see how these two areas interact. Also, this study can have real impact on addressing the current obesity issue in dogs and cats. Please see the following comments on the manuscript.

General comment

1. Classification of body condition score (BCS): One of my issues with BCS is that I cannot find sufficient evidence to support the cut-off values for underweight, normal weight, overweight and obesity. The problem, at least for me, is that when we investigate whether high BCS is associated with disease and a shorter lifespan, we pre-categorise then into “overweight” verse “non-overweight” instead of letting the data tell us which animals of which BCS are healthy or live longer and which are not. In cats, an increase in BSC from BCS 5 is associated with many diseases, but only cats reached BCS9 showed shorten lifespan (and the life span result is only based on one study). This has an implication on your analysis. For example, what is a BCS 6 should be considered normal weight than overweight, you would have misclassified your animals. Since you have quite a lot of animals and you will probably satisfy the assumption of a chi-square test, why not give it a try to analyse the data as it is without regrouping? Or with minimal regrouping. Or look for evidence to support your categorisation.

2. I think you have too much information to just run chi-square tests. Linear regression models would be more suitable, in my opinion, considering the richness of the information that you’ve gotten. Using linear regression modelling correctly (you probably want to transform your outcome(s) as they do not seem normally distributed), you could adjust for many factors that you consider to be associated with the outcome, and you will obtain more accurate and detailed results.

3. In my opinion, there are too few analyses with too much discussion that is not directly related to the current study (Lines 304-333). I understand the urge to discuss topics of our interests, but I think it’s important to be highly relevant as well. There can be more sophisticated analyses with more discussion on the results.

4. The questionnaire is not provided; thus it’s hard to evaluation the whole study. Please provide it.

5. The section of Results is particularly difficult to understand, compared to other sections. Please make some efforts to make it more readable.

6. Why was not the neglectful style investigated?

7. Did you estimate the sample size?

8. How was the survey advertised? Did it mention the aim of the study? Did it mention obesity and overweight? This is important as it is related to the selection of the target population and might introduce some biases.

9. Can the authors make some graphs to visualize the data for the readers? For example, the number and percentage of BCS and/or weight of dogs.

10. Please add a paragraph for limitations.

Specific comments

Line 42: What is the evidence of linking 15% to BCS6-7 and 30% to BCS8-9?

Lines 107-109: When was the survey period?

Lines 110-111: any other criteria? Such as age and fluency of Dutch? If not, please acknowledge it.

Lines 142-143: Not sure what it means exactly.

Lines 152-153: I would rephrase it.

Lines 164-165: I have to admit that I am not such a dog person, but over 70% of people walked their dogs for 1.5 hours or more seems a lot to me. Can you verify this in any way? Or support this by the literature.

Lines 167-168: It seems to me that some numbers got wrong here.

Lines 176-179 and 182-184: I am lost about the 7% of variation here. Please explain how this 7% was calculated, thanks. Variation of what?

Lines 185-187: It’s also quite unclear and confusing here.

Lines 209-212: I am not familiar with how the table 2 was presented with (residuals) and observed counts that had residual>2. Is it a common practice? The Residual>2 seems very arbitrary, especially you have larger and smaller expected values.

Lines 224-227: what does this 0.35 mean exactly? Also, it’s not just permissiveness explained the 12% of the variation (of what?) but also other covariates.

Lines 245-246: How about dogs? Don’t they have the ability to develop self-regulation/control during childhood and adolescence?

Lines 258-267: If I understand correctly, the authors were claiming that although in children, demandingness is associated with less obesity and permissiveness is associated with more obesity, only the later association holds in dogs. The former association was not observed because dogs remain dependent throughout its life. I am not sure this rationale works for me because, children, although being dependent on their parents, do have less probability to be obese when their parents have a demanding parenting style.

Lines 299-301: I actually disagree with the authors that owners with higher demandingness will judge loser the BCS. Any reference to support your claim? I would imagine that the permissive owners would instead be the population that does so.

Reviewer #2: The manuscript “Permissive parenting of the dog associates with dog overweight in a survey among 2,303 Dutch dog owners” deals with important topic and present interesting data. However, the way the authors perform the manuscript can and should be improved in order to increase the fluency and thus facilitate understanding.

Introduction

The first paragraph should be rewritten and reorganized since authors first indicate that obesity is a health problem, then describe % increase in BW in relation to denomination – obese overweight, then OH issues, and then again BW determination data etc. Therefore, the manuscript would greatly improve if the authors begin with the health problems, then state all related to BW and BCS and finally (this even could be separate paragraph) – OH.

Regarding the statement “dog overweight tends to coincide” is actually contradictory and different relations/no-relations have been described by different authors. This should be reflected in this manuscript as well.

The second paragraph actually describes the materials and results of the previous paper of the authors. This should be greatly shortened and only main findings and the very important data related to this manuscript should be included

The aims of the study should not contain references. Since if it is already described, there is no interest in this study. It would be preferable, that authors first indicate hypothesis and only then pass to aims.

Methods

How the authors controlled that there were no replicates?

Results

Please perfom table with the descriptive data of participants and the dogs. This will make it much easier to follow and understandable.

The number and % of lean/normal-weight/overweight-obese dogs should be also indicated. This data would be interesting to compare with the data reported for Denmark where also self-reported data obtained via questionnaires were used to evaluated the possible relationship between dogs and their owners (doi: 10.1038/s41598-018-31532-0.).

Discusion

Importantly, the limitations of the study should be reported. Furthermore, the authors should revise the discussion and leave only the most closely related to the data obtained in this study. For example, weight loss was not assessed in this study, but it occupies almost one page in the discussion.

6. PLOS authors have the option to publish the peer review history of their article (what does this mean?). If published, this will include your full peer review and any attached files.

Reviewer #1: No

Reviewer #2: No

---

## [Author Response · Author response to Decision Letter 0]

18 Feb 2020

Wageningen, February 16th 2020

Dear reviewers, 

We want to express our thanks for your constructive comments and feel we managed to process the comments effectively and/or address the issues in a logical and transparent way. We have revised the manuscript accordingly and please allow us to summarize how we responded to each point that was raised. 

Reviewer #1: 

1. Classification of body condition score (BCS)

We follow previous studies to classify the BCS into underweight, healthy-weight, overweight/obesity and list the references (line 52). In our study we do not determine whether high BCS associates with disease and a shorter lifespan. Merely we aim to study if and how BCS and dog-directed parenting associate (and we present information on our study approach in lines 145-158). The concern with regard to pre-categorising dogs into weight categories as to study health effects thus should not apply to our study. Furthermore, the implications of misclassifications are discussed in the Discussion (lines 284-298). We chose to analyse the data grouped in the often used underweight, healthy-weight, overweight/obesity classification as our participants dogs’ weights were not evenly spread over all BCS-categories, resulting in low counts in some categories and weakening the statistical power of the analysis. The reasons for merging BCS-categories is explained in the manuscript (lines 145-148). 

2. Chi-square tests or linear regression models

It was suggested to use more sophisticated statistics to explain variation in BCS scores (i.e. by linear regression with multiple explanatory variables), but this approach does not fit the purpose of the study and would result in the testing of an arbitrary set of explanatory variables. The latter as the survey was not designed to record variables that likely explain variation in BCS, but functioned to establish the relation between BCS and dog-directed parenting. Adding complexity to the statistical analysis would in this case be in discord with the purpose of the study, introduce subjectivity and make the results less straightforward. Thus, we opted for Chi-square tests and argumentation for this choice is presented in the manuscript in lines 148-155. 

3. Too much unrelated discussion

We removed lines 304-333 (previous submission) and generally shorted the Discussion section.

4. Questionnaire items

We added S3 Appendix, listing the parenting style questionnaire items.

5. Understandability of the Results section

We adapted the Results section, adding a Table and Figure, also responding to below ‘comment 9’.

6. Neglectful/uninvolved style

We added an explanation on not (being able to) study(ing) the uninvolved style (lines 127-129). 

7. Sample size

Sample sizes were not estimated through power calculations before making the survey available online and we did not predetermine a survey period. However, 95% of the data was collected in the first three months after starting the survey advertisement and we added this information to the manuscript (lines 111-113).

8. Survey advertisement, aim mentioning

We now provide details on how participants were informed on the study aim (lines 96-101). Details on study advertisement can be found in lines 90-92. 

9. Graph to visualize the data - BCS and/or weight of dogs

A graph on the dogs’ BCS was added to the Results section (Fig 1). 

10. Paragraph for limitations

We address the limitations of 1) our study population’s characteristics in lines 277-288, of 2) owner-self report as method in lines 288-292, of 3) underestimation in lines 292-302.

Specific comments

The specific comments were processed by deleting or adapting the lines that were insufficiently clear. Also, we added the study duration to the Methods section (lines 110-111), made a note on the Dutch language (line 96), added a reference on residuals (lines 158-161) and we commented on the presumably long dog walks of our participants in the Discussion section (lines 282-284). 

Reviewer #2: 

Introduction: 

We adapted the Introduction to create the suggested flow and removed the statements on One Health. Also, we shortened the section on dog-directed parenting styles and removed the details on the methods of the previous study. The aim was restated according to the feedback (lines 82-85). 

Methods: 

Details on replicates are now presented in lines 140-142. 

Results: 

A table replaces the previous text on the descriptive data of participants and their dogs (Table 1, starting line 174). Also, we moved the data on percentage and number of underweight, healthy-weight, overweight/obese dogs into this table. The suggestion for comparison with the European data was followed by adapting the Discussion in lines 284-288 and we thank the reviewer for pointing us to this reference.

Discussion: 

We shorted the Discussion section, removing all information on weight loss. We address the limitations of 1) our study population’s characteristics in lines 277-288, of 2) owner-self report as method in lines 288-292, of 3) underestimation in lines 292-302.

 With these adaptations made to our manuscript, we expect to have addressed all points raised.

---

## [Decision Letter · Decision Letter 1]

6 May 2020

PONE-D-19-29007R1

Permissive parenting of the dog associates with dog overweight in a survey among 2,303 Dutch dog owners

PLOS ONE

Dear Ir. Herwijnen,

Thank you for submitting your manuscript to PLOS ONE. After careful consideration, we feel that it has merit but does not fully meet PLOS ONE’s publication criteria as it currently stands. Therefore, we invite you to submit a revised version of the manuscript that addresses the points raised during the review process.

Whereas several of the initial shortcomings of the paper have been adequately dealt with in your revision, there are still problems in particular with how data were analysed and results are presented. Please see the reviewer reports below for detailed comments.

I want to draw your special attention to how to respond to reviewer comments. Especially when there are many and substantial comments, you really need to explicitly state, comment by comment, how you have considered it and which changes you have made in the paper in response to the comment. A more general response as the one you provided is making the reviewers' and editor's tasks unnecessarily difficult. 

I regret that it has taken a long time to get this answer to you. Given the disagreement over the appropriate analysis methods, we had to find a third reviewer for the statistics, and this was particularly challenging given the present circumstances of Covid-19 restrictions.

We would appreciate receiving your revised manuscript by Jun 20 2020 11:59PM. To enhance the reproducibility of your results, we recommend that if applicable you deposit your laboratory protocols in protocols.io, where a protocol can be assigned its own identifier (DOI) such that it can be cited independently in the future. For instructions see: http://journals.plos.org/plosone/s/submission-guidelines#loc-laboratory-protocols

We look forward to receiving your revised manuscript.

Kind regards,

I Anna S Olsson, Ph.D.

Academic Editor

PLOS ONE

Reviewers' comments:

Reviewer's Responses to Questions

**Comments to the Author**

1. If the authors have adequately addressed your comments raised in a previous round of review and you feel that this manuscript is now acceptable for publication, you may indicate that here to bypass the “Comments to the Author” section, enter your conflict of interest statement in the “Confidential to Editor” section, and submit your "Accept" recommendation.

Reviewer #1: (No Response)

Reviewer #2: All comments have been addressed

Reviewer #3: (No Response)

2. Is the manuscript technically sound, and do the data support the conclusions?

Reviewer #1: Partly

Reviewer #2: Yes

Reviewer #3: Partly

3. Has the statistical analysis been performed appropriately and rigorously? 

Reviewer #1: No

Reviewer #2: Yes

Reviewer #3: No

4. Have the authors made all data underlying the findings in their manuscript fully available?

Reviewer #1: Yes

Reviewer #2: Yes

Reviewer #3: Yes

5. Is the manuscript presented in an intelligible fashion and written in standard English?

Reviewer #1: Yes

Reviewer #2: Yes

Reviewer #3: Yes

6. Review Comments to the Author

Reviewer #1: Please see my comments as below.

Major points

1. The way that the authors responded to the reviewers’ comments was difficult for me to follow what you have changed during the last revision. It’s unclear whether the authors have addressed all the points raised by the reviewers.

2. The second comment that I made (Reviewer 1) was to run a multivariable linear model with the permissive style score as the outcome, which will answer the question that the authors intended to ask. Or you could run a multinomial logistic regression using the categories that you currently use in the Chi-square test. A big potential problem is that other variables such as sex, education level, walk time could even explain better your variation of the permissive style score than BCS, and you do have these pieces of information. Thus, I think this shouldn’t be ignored as what it is currently done in this manuscript.

Minor points

1. Lines 25-28: What is the hypothesis here?

2. Line 46: I am not sure about calling BCS measurement “subjective”. Yes, it’s rather subjective but it does have guidelines to be followed.

3. Lines 92-93: Is this an assumption or you provided guidelines and ensured that it was like this?

4. Lines 102-103: What did you do to make sure (a) only taken once and (b) not causing psychologically burdening?

5. I think Figure 1 has a better way to be presented. Also, this information was given in the table, and they were inconsistent.

Reviewer #2: The authors performed modifications as suggested. However, it would be appreciated if the authors would have copy/paste the indications of the reviewers in their response letter directly and not summarizing. This would have facilitated the re-revision. In actual format, I had to look for the reviewers original indications and compare them with the data reported by authors and, therefore, it took me much more time to make the revision.

Reviewer #3: 1) The authors have some discussion of replication in the data and missing values, but no quantitative assessments.

Please generate a CONSORT-style diagram for the study. This can notate the numbers obtained from various sources, the number excluded from the study due to duplication or any other issues (such as analysis inclusion and exclusion criteria), and the number used for various analyses.

Please include a discussion at the beginning of the results section about the number and percentage of missing values. Or, at a minimum, in a supplementary data quality appendix (but, even then, there should be at least a brief mention in the text).

2) The authors do note some issues relating to the generalizability of their results. One major issue that is still undiscussed is that the sample was completely self-selected. If possible, perhaps the demographics of the sample can be compared with known population results. In any case, the authors need to spend some time assessing how this aspect of the study may affect the interpretation of the results.

3) Table 1 is bizarrely formatted. Please use a standardized table format. Given that the authors present results by parenting style, perhaps the table could be formatted with those styles across the top, along with a total column.

4) The authors split the data ambiguously. Please clearly indicate whether 0-19 means 0 to 19 inclusive or exclusive of 19. Present intervals as either 0-19, 20-25 or as 0-18, 19-24 as appropriate.

Also, the last category should be 35-100, presumably. Altogether, it would be better if the authors had an a priori split in mind rather than an a posteriori split on sample quantiles. The authors' motivation of the methodology in the chi-squared test is somewhat dubious, but there is no need to argue the point. However, I notice that even just splitting the PERM data into [0, 10], (10, 20], ..., (90, 100] and testing it versus the BCS yields an equally statistically significant result. The important point is to interpret the actual distribution changes from quartile to quartile or intervals.

5) The authors tabulate and present statistical results only for permissive parenting stye. Please present results for all three styles considered. This can be done in three sections in the table:

Body Condition Score groups across the columns.

* Permissive

** Score 0-19

** Score 19-25

** Score 25-35

** Score 35-100

* Authoritative

** Score 0-19

** Score 19-25

** Score 25-35

** Score 35-100

* Authoritarian

** Score 0-19

** Score 19-25

** Score 25-35

** Score 35-100

The p-value may be presented in the last column.

6) Was there a protocol prepared for this study? If so, what were the preplanned statistical analyses? Were they performed? If so, they need to be reported.

7) Figure 1 really seems like a low-effort presentation. How about a more useful presentation, such as a histogram of the actual scores? Or, histograms for each parenting style?

8) Are the data collected of no other value? What about the relationships among the other variables? Are any of those of any interest?

Is there any reason these should not at least be tabulated using mean, standard deviation, minimum, and maximum for each of the groups?

7. PLOS authors have the option to publish the peer review history of their article (what does this mean?). If published, this will include your full peer review and any attached files.

Reviewer #1: No

Reviewer #2: No

Reviewer #3: No

---

## [Author Response · Author response to Decision Letter 1]

19 Jun 2020

Reviewer #1

1. The way that the authors responded to the reviewers’ comments was difficult for me to follow what you have changed during the last revision. It’s unclear whether the authors have addressed all the points raised by the reviewers.

In our previous response letter, we addressed all the points raised by the reviewers. We chose to present our response concise as to facilitate reading, but regret to be informed that this was not helpful. In this response to the reviewers, we address the points one by one. 

2. The second comment that I made (Reviewer 1) was to run a multivariable linear model with the permissive style score as the outcome, which will answer the question that the authors intended to ask. Or you could run a multinomial logistic regression using the categories that you currently use in the Chi-square test. A big potential problem is that other variables such as sex, education level, walk time could even explain better your variation of the permissive style score than BCS, and you do have these pieces of information. Thus, I think this shouldn’t be ignored as what it is currently done in this manuscript.

We thank the reviewer for the suggestions. There is now a third paragraph in the Results section (‘Modifiers of relationships between parenting style and overweight/obesity’). This paragraph indicates how other possibly weight-explaining variables may affect the found association of our interest, between permissive parenting and a dog’s body weight (from line 263 onwards).

3. Minor points

3.1. Lines 25-28: What is the hypothesis here? 

We have sharpened the hypothesis to more clearly indicate that ‘We predicted that permissive dog-directed parenting (…) associates with a dog being overweight (…)’ (lines 26-27).

3.2. Line 46: I am not sure about calling BCS measurement “subjective”. Yes, it’s rather subjective but it does have guidelines to be followed. 

We have removed the word ‘subjective’ (line 50). 

3.3. Lines 92-93: Is this an assumption or you provided guidelines and ensured that it was like this?

The sentence on social pressure was adapted to improve understanding, and now reads: ‘For responding to the survey, the participants were not invited to a research location, but presumedly responded to our web-based survey on a private computer, at home or elsewhere, so without social pressure to provide certain answers.’ (lines 97-99). 

3.4 Lines 102-103: What did you do to make sure (a) only taken once and (b) not causing psychologically burdening? 

The sentence was adapted to improve understanding and now reads ‘The survey was taken once, we did not contact the participants for a second survey, did not include questions that were psychologically burdening, that is psychologically difficult to bear such as on mental health status, meaning it did not interfere significantly with normal daily life.’ (lines 108-110).

3.5. I think Figure 1 has a better way to be presented. Also, this information was given in the table, and they were inconsistent. 

We have removed Figure 1 as to prevent the confusion caused by the rounding differences between this Figure and the Table. We have contemplated how to process the varying feedback from all reviewers. We now present a table on the general statistics of Body Condition Scores (BCS) and parenting style (starting at line 231) and the other descriptive data on the participants and their dogs as text (lines 215-229). We feel this is the most transparent and concise manner in which to present the data for the type of study we did.

Reviewer #2

The authors performed modifications as suggested. However, it would be appreciated if the authors would have copy/paste the indications of the reviewers in their response letter directly and not summarizing.

In our previous response letter we addressed all the points raised by the reviewers. We chose to present our response concise as to facilitate reading, but regret to be informed that this was not helpful. In this response to the reviewers, we address the points one by one and we have copied the indications of the reviewers above each comment, as per request. 

Reviewer #3

1) The authors have some discussion of replication in the data and missing values, but no quantitative assessments. Please generate a CONSORT-style diagram for the study. This can notate the numbers obtained from various sources, the number excluded from the study due to duplication or any other issues (such as analysis inclusion and exclusion criteria), and the number used for various analyses. Please include a discussion at the beginning of the results section about the number and percentage of missing values. Or, at a minimum, in a supplementary data quality appendix (but, even then, there should be at least a brief mention in the text).

We thank the reviewer for pointing out that we did not state clearly enough that we depended on one study sample only, for all statistical analyses, but that answering of questions in the survey was not mandatory and as a consequence unanswered questions are presented as missing values. To improve this, we have added a section indicating how our sample of 2,303 surveys originated. We now clearly state that this sample was used for all statistical analyses (lines 154-159). In line with the reviewer’s comment, we also present the number of surveys excluded due to a missing value on body condition scores at the start of the results section (lines 210-211). Please note that our study does not regard a trial. Entering a consort style diagram may wrongly suggest a higher level of research than our present survey-based study. (We are merely aiming to gather a first insight on a possible relationship between parenting styles and body weight based on associative, but not causal, evidence.)

2) The authors do note some issues relating to the generalizability of their results. One major issue that is still undiscussed is that the sample was completely self-selected. If possible, perhaps the demographics of the sample can be compared with known population results. In any case, the authors need to spend some time assessing how this aspect of the study may affect the interpretation of the results. 

We have adapted lines 381 to 388 incorporating a discussion of the matter of self-selection into the already discussed matter of study sample composition and the implications thereof. Our study group has previously attempted to obtain reliable information on the population of Dutch dog owners, e.g. by contacting Statistics Netherlands (CBS), but were not able to find useful facts. This is now noted in the Discussion (lines 389-391): ‘We searched for demographics of the Dutch dog owner population but were unable to find statistics that could be used for comparison with our study group.’

3) Table 1 is bizarrely formatted. Please use a standardized table format. Given that the authors present results by parenting style, perhaps the table could be formatted with those styles across the top, along with a total column.

We agree that, for our study type, a table is not the optimal format to present the participants’ information, but followed the advice kindly provided by Reviewer #2 from the former peer review feedback round. We have contemplated how to process the varying feedback from all reviewers. We now present a table on the general statistics of Body Condition Scores (BCS) and parenting style (starting at line 231) and the other descriptive data on the participants and their dogs as text (lines 215-229). We feel this is the most transparent and concise manner in which to present the data for the type of study we did.

4) The authors split the data ambiguously. Please clearly indicate whether 0-19 means 0 to 19 inclusive or exclusive of 19. Present intervals as either 0-19, 20-25 or as 0-18, 19-24 as appropriate. Also, the last category should be 35-100, presumably. Altogether, it would be better if the authors had an a priori split in mind rather than an a posteriori split on sample quantiles. The authors' motivation of the methodology in the chi-squared test is somewhat dubious, but there is no need to argue the point. However, I notice that even just splitting the PERM data into [0, 10], (10, 20], ..., (90, 100] and testing it versus the BCS yields an equally statistically significant result. The important point is to interpret the actual distribution changes from quartile to quartile or intervals.

Data ranges with ‘-‘ were inclusive and we now make this clear in the document for instance by adding ‘inclusive’ at the first time this letter sign ‘-‘ was used for this purpose (line 137). We present the quartiles of parenting style scores over the true data quartiles, that is based on our data’s distribution. This implies that the threshold values in our dataset determine the demarcation of the quartiles. For permissive parenting the range was 0-92% (hence using 100% as outer range we feel would be incorrect). We have now indicated the approach more clearly in the methods section (lines 167-168).

5) The authors tabulate and present statistical results only for permissive parenting style. Please present results for all three styles considered. This can be done in three sections in the table:

Body Condition Score groups across the columns.

* Permissive

** Score 0-19

** Score 19-25

** Score 25-35

** Score 35-100

* Authoritative

** Score 0-19

** Score 19-25

** Score 25-35

** Score 35-100

* Authoritarian

** Score 0-19

** Score 19-25

** Score 25-35

** Score 35-100

The p-value may be presented in the last column.

We agree that all data should be available and present the requested overviews in S2 Table. We value the focus on the permissive style in our article and moved the table that is now in S2 from the main text after previous feedback rounds with pre-readers of the article.

6) Was there a protocol prepared for this study? If so, what were the preplanned statistical analyses? Were they performed? If so, they need to be reported.

We did not prepare a protocol or work with pre-planned statistical analyses. We aimed to investigate associations between dog-directed parenting styles and body condition scores in dogs, using a robust, basic, statistical method as to lay a foundation for future more hypotheses-driven experimental studies. We have stressed the need for more hypotheses-driven experimental studies in lines 412-413 of the Discussion section.

7) Figure 1 really seems like a low-effort presentation. How about a more useful presentation, such as a histogram of the actual scores? Or, histograms for each parenting style?

We have contemplated how to process the varying feedback from all reviewers. We now present a table on the general statistics of Body Condition Scores (BCS) and parenting style (starting at line 231) and the other descriptive data on the participants and their dogs as text (lines 215-229). We feel this is the most transparent and concise manner in which to present the data for the type of study we did.

8) Are the data collected of no other value? What about the relationships among the other variables? Are any of those of any interest? Is there any reason these should not at least be tabulated using mean, standard deviation, minimum, and maximum for each of the groups?

Our study set up was not designed to explain variation in BCS scores or relationships between a larger range of variables, but functioned to establish the relation between BCS and dog-directed parenting. Adding complexity to the statistical analysis would in this case be in discord with the purpose of the study, introduce subjectivity and make the results less straightforward. However, we see the value in indicating how other variables and relationships between these and weight/permissiveness may exist. Therefore, we added a third paragraph to the Results section, starting at line 263. This paragraph indicates how other possibly weight-explaining variables may affect the found association of our interest, between permissive parenting and a dog’s body weight. 

As participants dogs’ weights were not evenly spread over all body condition score categories, resulting in low counts in some categories, it is more fitting to describe the data with medians and ranges than with means and standard deviations. We have now indicated this in lines 172-173.

---

## [Decision Letter · Decision Letter 2]

23 Jul 2020

PONE-D-19-29007R2

Permissive parenting of the dog associates with dog overweight in a survey among 2,303 Dutch dog owners

PLOS ONE

Dear Dr. Herwijnen,

Thank you for submitting your manuscript to PLOS ONE. After careful consideration, we feel that it has merit but does not fully meet PLOS ONE’s publication criteria as it currently stands. Therefore, we invite you to submit a revised version of the manuscript that addresses the points raised during the review process.

I recognize that your previous revision has addressed the reviewer comments in a satisfactory way. After carefully reading the paper, I have a few minor additional remarks:

(please note that the line numbers refer to the track-changes version)

Line 101 Social desirability bias is still at play in anonymous online surveys, so please reword "so without social pressure" to "hence minimizing social pressure"

Line 112 Replace "it did not interfere" with "it was unlikely to interfer"

Lines 342-343 This sentence is difficult to understand. Please consider rewording to "Weight management of the dog could be one expression of dog-directed parenting, similarly to ...."

Line 367 Please replace "which lacks in permissive parenting" with "which is low in permissive parenting" (because demandingness is low rather than absent in permissive parenting, right?)

We look forward to receiving your revised manuscript.

Kind regards,

I Anna S Olsson, Ph.D.

Academic Editor

PLOS ONE

Reviewers' comments:

Reviewer's Responses to Questions

**Comments to the Author**

1. If the authors have adequately addressed your comments raised in a previous round of review and you feel that this manuscript is now acceptable for publication, you may indicate that here to bypass the “Comments to the Author” section, enter your conflict of interest statement in the “Confidential to Editor” section, and submit your "Accept" recommendation.

Reviewer #3: All comments have been addressed

2. Is the manuscript technically sound, and do the data support the conclusions?

Reviewer #3: (No Response)

3. Has the statistical analysis been performed appropriately and rigorously? 

Reviewer #3: (No Response)

4. Have the authors made all data underlying the findings in their manuscript fully available?

Reviewer #3: (No Response)

5. Is the manuscript presented in an intelligible fashion and written in standard English?

Reviewer #3: (No Response)

6. Review Comments to the Author

Reviewer #3: The authors have made a variety of changes to address most of the critique of the statistical methods used. While there is still room for improvement, in my judgement it seems unlikely that making those improvements would change the tenor of the basic results. The data are of interest, though likely subject to unknown levels of sampling bias, so they should be published. Presumably the original data will be available as part of the publication, so that other researchers may apply other methodology as desired.

I would like to recommend that for future trials in this area, that it would be best to set out a protocol which details the data that will be collected as well as the statistical methods that will be used to analyze the data. My recommendation is usually to perform a pilot study to collect a small set of real data to be analyzed --- however, it is also useful to mock up data and carry out an analysis on the mock data. These types of preparation will pay off inordinately in comparison to the up-front effort entailed. Though the authors note that the study is not a trial per se, that does not diminish the value of data collected in a well-planned survey. My feeling is that many of the authors' issues with regard to the treatment of the data arose from the somewhat ad hoc and reactive analysis that was performed. It is difficult to change course in midstream sometimes, so it is best to chart an accurate course at first.

7. PLOS authors have the option to publish the peer review history of their article (what does this mean?). If published, this will include your full peer review and any attached files.

Reviewer #3: No

---

## [Author Response · Author response to Decision Letter 2]

26 Jul 2020

Thank you for the minor additional remarks.

Line 101 Social desirability bias is still at play in anonymous online surveys, so please reword "so

without social pressure" to "hence minimizing social pressure"

Adjusted accordingly.

Line 112 Replace "it did not interfere" with "it was unlikely to interfere"

Adjusted accordingly.

Lines 342-343 This sentence is difficult to understand. Please consider rewording to "Weight

management of the dog could be one expression of dog-directed parenting, similarly to ...."

Adjusted accordingly.

Line 367 Please replace "which lacks in permissive parenting" with "which is low in permissive

parenting" (because demandingness is low rather than absent in permissive parenting, right?)

Adjusted accordingly.

We also thank the reviewer for the final comment, recommending to set out a protocol up front and to perform a small pilot study first. We will take advantage of this advice when preparing future studies.

---

## [Editor Report · Decision Letter 3]

28 Jul 2020

Permissive parenting of the dog associates with dog overweight in a survey among 2,303 Dutch dog owners

PONE-D-19-29007R3

Dear Dr. Herwijnen,

We’re pleased to inform you that your manuscript has been judged scientifically suitable for publication and will be formally accepted for publication once it meets all outstanding technical requirements.

Kind regards,

I Anna S Olsson, Ph.D.

Academic Editor

PLOS ONE
---

## [Editor Report · Acceptance letter]

31 Jul 2020

PONE-D-19-29007R3 

 Permissive parenting of the dog associates with dog overweight in a survey among 2,303 Dutch dog owners

Dear Dr. Herwijnen:

I'm pleased to inform you that your manuscript has been deemed suitable for publication in PLOS ONE. Congratulations! Your manuscript is now with our production department. 

Kind regards, 

on behalf of

Dr. I Anna S Olsson 

Academic Editor

PLOS ONE